# Progression-Free and Overall Survival of First-Line Treatments for Advanced Renal Cell Carcinoma: Indirect Comparison of Six Combination Regimens

**DOI:** 10.3390/cancers15072029

**Published:** 2023-03-29

**Authors:** Andrea Ossato, Daniele Mengato, Marco Chiumente, Andrea Messori, Vera Damuzzo

**Affiliations:** 1Department of Pharmaceutical and Pharmacological Sciences, University of Padua, 35131 Padua, Italy; 2Hospital Pharmacy Department, University Hospital of Padua, 35128 Padua, Italy; daniele.mengato@gmail.com; 3Italian Society of Clinical Pharmacy and Therapeutics (SIFaCT), 10123 Turin, Italy; marco.chiumente@sifact.it; 4HTA Unit, Regional Health Service, 50139 Florence, Italy; andrea.messori.it@gmail.com; 5Hospital Pharmacy, Vittorio Veneto Hospital, 31029 Vittorio Veneto, Italy; vdamuzzo@gmail.com

**Keywords:** indirect comparison, Shiny method, reconstructed individual patient data, overall survival, renal cell carcinoma, combination

## Abstract

**Simple Summary:**

Recently, numerous treatments sharing similar mechanisms of action have been approved for advanced renal cell carcinoma. These combinations prolong survival compared to sunitinib, which was previously considered the standard of care in this context. Head-to-head comparisons between these innovative treatments are not available, but this information is needed to guide medical oncologists’ choices. To compare these combination therapies with one another and with sunitinib, our study used an innovative method (the Shiny method) that reconstructs individual patient data from published clinical trials. Using this approach, we demonstrated that pembrolizumab + lenvatinib is the most effective treatment in terms of progression-free survival (PFS) and overall survival (OS). Pembrolizumab + axitinib, nivolumab + cabozantinib and nivolumab + ipilimumab were similar in terms of PFS and superior to sunitinib, but pembrolizumab + axitinib also demonstrated a better OS. Our subgroup analysis showed that in favorable-risk patients, combination therapies showed no significant advantage over sunitinib, while in intermediate-poor risk patients, both pembrolizumab + axitinib and nivolumab + ipilimumab improved OS compared to sunitinib.

**Abstract:**

Background: Recently, numerous combination therapies based on immune checkpoint inhibitors (ICI) and vascular endothelial growth factor (VEGF) inhibitors have been proposed as first-line treatments for advanced renal cell carcinoma (aRCC). Our study aimed to compare the efficacy of these combination regimens by the application of an innovative method that reconstructs individual patient data. Methods: Six phase III studies describing different combination regimens for aRCC were selected. Individual patient data were reconstructed from Kaplan–Meier (KM) curves through the “Shiny method”. Overall survival (OS) and progression-free survival (PFS) were compared among combination treatments and sunitinib. Results were summarized as multi-treatment KM curves. Standard statistical testing was used, including hazard ratio and likelihood ratio tests for heterogeneity. Results: In the overall population of aRCC patients, pembrolizumab + lenvatinib showed the longest median PFS and was expected to determine the longest OS. Pembrolizumab + axitinib, nivolumab + cabozantinib and nivolumab + ipilimumab were similar in terms of PFS, but pembrolizumab + axitinib also demonstrated a better OS. Our subgroup analysis showed that sunitinib is still a valuable option, whereas, in intermediate-poor risk patients, pembrolizumab + axitinib and nivolumab + ipilimumab significantly improve OS compared to sunitinib. Conclusion: The Shiny method allowed us to perform all head-to-head indirect comparisons between these agents in a context in which “real” comparative trials have not been performed.

## 1. Introduction

Over the past decades, systemic treatment for advanced renal cell carcinoma (aRCC) has considerably improved along with the development of new pharmacological targets. The main advancement in this setting has been the development of monoclonal antibodies and multitarget tyrosine-kinase inhibitors (TKI) that inhibit tumor growth and angiogenesis through the vascular endothelial growth factor (VEGF) pathway (e.g., bevacizumab, axitinib, lenvatinib, cabozantinib and sunitinib) [1].

Later on, inhibitors of immune checkpoints (ICIs), such as programmed cell death receptor-1 and its ligand (PD-1: nivolumab, pembrolizumab, PDL-1: atezolizumab and avelumab) and cytotoxic T lymphocyte-associated molecule-4 (CTLA-4; e.g., ipilimumab) have also demonstrated efficacy in aRCC [2,3].

While ICI were initially used as monotherapy after TKI, further evidence showed that combining these agents improved efficacy with an acceptable safety profile [3]. To date, combination regimens to treat aRCC include ICI/TKI (avelumab + axitinib, pembrolizumab + lenvatinib, nivolumab + cabozantinib and pembrolizumab + axitinib), ICI/anti-VEGF mAb (atezolizumab + bevacizumab) and a combination of two ICI (nivolumab + ipilimumab) [4,5].

In this ever-changing scenario, comparisons between these innovative treatments are essential to drivinge physician and regulatory authorities’ decisions; however, randomized clinical trials (RCTs) have tested the efficacy of combination treatments only against sunitinib. On the other hand, in the field of methods for survival analysis, a new artificial intelligence technique (called the “Shiny method”) has been developed and is increasingly used to reconstruct individual patient data from Kaplan-Meier (KM) curves and to generate cross-trial comparisons for which RCTs are lacking [6,7,8].

The Shiny method, also known as IPDfromKM, is an innovative tool of survival analysis in which software based on artificial intelligence reconstructs individual patient data. These reconstructed patients represent a new form of original clinical material; in particular, these reconstructed patients are suitable to perform indirect comparisons and consequently determine the place in the therapy of individual agents. In recent papers, we described the method and numerous experiences of application, especially in the field of oncology [9]. These analyses are helpful in providing an overview of new treatments, ranking their effectiveness based on indirect comparisons, and assessing equivalence from a regulatory viewpoint.

In this study, we applied the “Shiny method” to compare the efficacy of the main ICI-based combination regimens for aRCC, considering sunitinib as the standard of care (SOC).

## 2. Materials and Methods

### 2.1. Literature Search

First, we searched PubMed, Cochrane Library and the ClinicalTrials.gov databases to identify randomized controlled trials (RCTs) eligible for our analysis (last query on 30 October 2022). The search term was “[[advanced] or [metastatic]] and [[renal cell carcinoma] or [RCC]]”. The time interval was January 2019 to the present date. The selection of articles from our literature search was based on the PRISMA algorithm [10].

The main inclusion criteria were: (a) previously untreated adult patients with aRCC; (b) phase III trial; (c) PFS or OS endpoint; (d) results reported as a KM curve. For each curve of included trials, we collected the number of enrolled patients and the number of events (progression or death for PFS, death for OS). To avoid duplicate inclusion of patients of the same trial, we considered only the most recent publication.

### 2.2. Reconstruction of Individual Patient Data

We reconstructed patient-level data from the KM curves of treatment and control arms of each trial using the “Shiny method” [6,7,8]. For this purpose, the KM curves were digitized using Webplotdigitizer (version 4.5 online; https://apps.automeris.io/wpd/ (accessed on 15 February 2023)); then, the x-vs-y data points were input into the “Reconstruct individual patient data from Kaplan-Meier survival curve” function of the Shiny software (version: 1.2.2.0 online; last update: 1 April 2021); the total number of patients and events were input as well. In this way, we generated the reconstructed individual patient data for each arm of included RCTs. The reconstructed data of the patients are stored in archives containing the following information: (a) date of enrollment and last follow-up; the observation period of each patient results from the difference between these two dates. (b) patient outcome on the last follow-up date (alive, dead or censored).

As sunitinib was considered the SOC against which combination therapies were compared, patients receiving sunitinib monotherapy were pooled to form a single control group composed of patients from the control group of the 6 trials, namely IMmotion151 (*n* = 461), JAVELIN Renal 101 (*n* = 444), CheckMate 214 (*n* = 546), CLEAR (*n* = 357), CheckMate 9ER (*n* = 328) and KEYNOTE-426 (*n* = 429) trials [11,12,13,14,15,16].

### 2.3. Statistical Analysis

For each combination treatment, median PFS and OS were determined from reconstructed data and compared to sunitinib using Cox statistics for time-to-event end-points. We reported results as hazard ratio (HR) with a 95% confidence interval (95%CI). Indirect comparisons between treatments (in all head-to-head combinations) have been performed using the Cox model under the R-platform. Heterogeneity across control groups of different RCTs was quantified according to the likelihood ratio test and the concordance test. Statistical analyses were performed using the “survival” package under the R-platform (version 4.2.1) and SygmaPlot (version 13) software.

## 3. Results

### 3.1. Included Trials and Application of the Shiny Method

Six trials met the criteria for inclusion in our analysis (see Figure 1 for the PRISMA flowchart and Table 1 for RCT characteristics).

In our analysis of PFS based on these trials, 12 patient cohorts, along with their respective information on progressions, were reconstructed from the original Kaplan-Meier curves through the Shiny method. Then, the Kaplan-Meier PFS curves from these reconstructed patients were plotted individually and reported in a single multi-treatment graph (Figure 2A), which is the typical result generated by the Shiny method. In this graph, the 6 cohorts treated with sunitinib were pooled into a single cohort, thus generating a total of 7 curves. The same analysis was then performed for OS. The latter results are reported in Figure 2B. Detailed results of all head-to-head comparisons are reported in Table 2 and Appendix A.

### 3.2. Progression-Free Survival: Indirect Comparisons of the Six Combination Treatments Plus Sunitinib with One Another

In analyzing our results on PFS, the six combination treatments, along with sunitinib monotherapy, were ranked as follows:pembrolizumab + lenvatinib: median PFS 24.52 months (95%CI 22.02 to 29.5); this combination treatment is significantly better than all other combination treatments, in particular of nivolumab + cabozantinib, which ranks second, with HR = 0.714 (95%CI 0.574 to 0.889);nivolumab + cabozantinib: median PFS 17.12 months (13.01 to 20.2);pembrolizumab + axitinib: median PFS 16.99 months (14.21 to 21.1);nivolumab + ipilimumab: median PFS 14.29 months (12.02 to 18.8); these three combinations produced similar gains of survival with respect to sunitinib (see Appendix A), with no difference in the head-to-head comparisons; in contrast, they were significantly superior to atezolizumab + bevacizumab and numerically superior to avelumab + axitinib;avelumab + axitinib: median PFS 13.85 months (95%CI 10.99 to 16.7) was inferior to all other combinations except atezolizumab + bevacizumab (HR = 0.817, 95%CI 0.678 to 0.985);atezolizumab + bevacizumab: median PFS 11.45 months (95%CI 9.87 to 13.5) was significantly inferior to all other treatments, and it does not produce any significant gain of function compared to sunitinib (HR = 0.9822, 95%CI 0.8637 to 1.117).sunitinib (6 control arms pooled together): median PFS 9.92 months (95%CI 9.53 to 10.09) was inferior to 5 out of six combination treatments, with the exception of atezolizumab + bevacizumab.

Forest plots of HRs with 95%CI for PFS of the six trials against pooled sunitinib controls are shown in Appendix A.

### 3.3. Overall Survival: Indirect Comparisons of the 6 Combination Treatments Plus Sunitinib with One Another

In analyzing our results on OS, the six combination treatments, along with sunitinib monotherapy, were ranked as follows:pembrolizumab + lenvatinib: median OS not reached, but this treatment is expected to rank first based on the HR analysis vs. sunitinib (HR = 0.57, 95%CI 0.45 to 0.72); in the head-to head comparison, pembrolizumab + lenvatinib was numerically superior to both pembrolizumab + axitinib (HR = 0.868, 95%CI 0.634 to 1.189) and nivolumab + ipilimumab (HR = 0.781, 95%CI 0.592 to 1.031) but significantly better than other combinations.pembrolizumab + axitinib: median OS not reached, but this treatment is expected to rank second based on the HR analysis vs. sunitinib (HR = 0.6601, 95%CI 0.5369 to 0.8116); in the head-to head comparison, pembrolizumab + axitinib was numerically superior to nivolumab + ipilimumab, nivolumab + cabozantinib and avelumab + axitinib and significantly superior to atezolizumab + bevacizumab;nivolumab + ipilimumab: median OS 59.2 months (48 to 70.3). This treatment was numerically superior to nivolumab + cabozantinib and avelumab + axitinib;nivolumab + cabozantinib: median OS 37.9 months (95%CI 36.3 to NA). This combination failed to demonstrate any significant advantage compared to other treatments but it was numerically superior to atezolizumab + bevacizumab and almost similar to avelumab + axitinib;avelumab + axitinib: median OS not reached. This treatment was numerically superior only to atezolizumab + bevacizumab and sunitinib monotherapy;sunitinib (6 control arms pooled together): median OS 41.2 months (95%CI 35.6 to 46.5) was inferior to 5 out of six combination treatments, with the exception of atezolizumab + bevacizumab;atezolizumab + bevacizumab: median OS 35.5 months (95%CI 28.9 to NA). This treatment was significantly worse than most of the comparators, including sunitinib monotherapy. The HR analysis vs. sunitinib failed to demonstrate a significant gain in survival (HR = 1.0818, 95%CI 0.9148 to 1.2794).

Forest plots of HRs with 95%CI for OS of the six trials against pooled sunitinib controls are shown in Appendix A. Detailed results of all head-to-head comparisons are reported in Table 3 and Appendix A.

### 3.4. Heterogeneity Analysis on Sunitinib Monotherapy Curves

Figure 3A,B shows the curves of PFS and OS obtained from the 6 control groups treated with sunitinib monotherapy that were compared for heterogeneity assessment. As controls should behave in a similar way, if heterogeneity is present, this may be due to differences in baseline patients’ characteristics across the 6 trials. The results indicated that heterogeneity was present among the six trials both according to the likelihood ratio test and the concordance test applied to PFS (likelihood test = 50.06 with 5 df; *p* < 0.001; concordance, 0.552, SE = 0.007) and OS (likelihood test = 15.36 with 5 df; *p* = 0.009; concordance, 0.534, SE = 0.01. The presence of heterogeneity is driven mainly by the nivolumab + ipilimumab and lenvatinib + pembrolizumab RCT control arm, respectively, in the PFS and OS analysis.

In the case of PFS, there is no longer a significant heterogeneity when the nivolumab + ipilimumab control group is left out (likelihood ratio test = 8.86 on 4 df, *p* = 0.06); this demonstrates that in this case, the heterogeneity is driven mainly by this control arm.

In contrast, in the case of OS analysis, there is still significant heterogeneity after excluding the nivolumab + ipilimumab control group (likelihood ratio test = 14.6 on 4 df, *p* = 0.006). In this case, the lenvatinib + pembrolizumab control arm performed significantly better than the others, and so the heterogeneity was driven by this curve in the OS analysis. As expected, the likelihood ratio test after excluding the lenvatinib + pembrolizumab control group was not significant (likelihood ratio test = 3.56 on 4 df, *p* = 0.5).

### 3.5. Subgroup Analysis: Overall Survival in Favourable vs. Intermediate/Poor Risk Patients

In patients treated with nivolumab + ipilimumab and pembrolizumab + axitinib, OS’s KM curves were available for patients with favourable versus intermediate/poor risk.

Multi-treatment KM curves are depicted in Figure 4A,B.

In patients with favourable risk, median OS values of nivolumab + ipilimumab and pembrolizumab + axitinib do not significantly differ from sunitinib group (HR = 1.026, 95%CI 0.6398 to 1.4866 for nivolumab + ipilimumab and HR = 1.024, 95%CI 0.5500 to 1.735 for pembrolizumab + lenvatinib axitinib vs. sunitinib). The head-to-head comparisons do not show any difference between the two combination treatments.

In the intermediate-poor risk group, sunitinib performed significantly worse than combination regimens (median OS 29.4 months, 95%CI 25.4–34.3). Nivolumab + ipilimumab (HR = 1.437, 95%CI 0.5872 to 0.8247) improve OS compared to sunitinib monotherapy similarly to pembrolizumab + axitinib (HR = 1.579, 95%CI 0.5011 to 0.8001). The head-to-head comparison between these combinations do not show any difference (HR = 0.91, 95%CI 0.682 to 1.215).

## 4. Discussion

The present study investigated the main first-line combination treatments for aRCC by application of an innovative method of indirect comparison of survival data, the “Shiny method”. Our choice to employ this approach to conduct these indirect comparisons, not a standard network meta-analysis [17,18,19], was because this method offers some advantages, such as considering the length of the follow-up of the trials and managing in a more precise way the variance of the model. The fact that the Shiny method evaluates not only the number of events but also the time of their occurrence is particularly important in a context like the one studied in here, where the follow-up lengths were considerably different across the agents investigated, and some differences were substantial. For example, some treatments (such as nivolumab + ipilimumab or sunitinib monotherapy and, to a lesser extent, lenvatinib + pembrolizumab and nivolumab + cabozantinib) had a particularly long follow-up as opposed to atezolizumab + bevacizumab, the follow-up of which was short. Consequently, some event rates were influenced by the length of the follow-up; in this framework, the rankings estimated by the Shiny method have an advantage in that they accounted for this important factor.

In recent years, the Shiny method has increasingly been used not only in oncology but also in other areas of of therapeutics, such as medical devices [20], surgery [21,22] and also in patients with COVID-19 [23]. However, oncology and oncohematology remain the two main areas of application, where the Shiny method represents a quite simple alternative to network meta-analysis [8,9,24,25].

Although combination regimens for aRCC target similar pathways, our results indicate that their efficacy is not equal. Median PFS and OS obtained with atezolizumab + bevacizumab did not differ significantly from that obtained with sunitinib, and so this result does not yet justify its use. In contrast, several combinations of TKI + ICI demonstrated a significant advantage, in terms of both PFS and OS, when compared to sunitinib monotherapy, in particular pembrolizumab + lenvatinib. Long-term follow-up will tell whether this therapy will confirm its superiority compared to the other combinations.

Pembrolizumab + axitinib, nivolumab + cabozantinib and nivolumab + ipilimumab were similar in terms of PFS but pembrolizumab + axitinib demonstrated a better OS.

In subgroup analysis, patients with favourable risk obtained excellent results in terms of OS also with sunitinib monotherapy, and so combination therapy in these patients seems to add little OS benefit. In contrast, the intermediate/poor risk group of patients seems to benefit mostly from combination therapy, though we found no difference between nivolumab + ipilimumab vs. pembrolizumab + axitinib. In intermediate/poor risk patients, no separate KM curves were reported for patients with intermediate/low risk treated with pembrolizumab + lenvatinib, and this prevented us from applying the Shiny method to test the superiority observed by Bosma et al. in their meta-analysis [17].

Although the criteria for patient selection were similar among the six studies, our heterogeneity analysis of control arms demonstrated a significantly better PFS and OS in the controls of the CheckMate214 trial (the one testing nivolumab + ipilimumab) compared with the other control arms. Hence, this raises the possibility that the remarkable survival found in the active arm of CheckMate214 depends on the favorable characteristics of the patients and is therefore an overestimate. To further assess this issue, we compared in more detail the inclusion criteria and the characteristics of patients in the selected RCTs, but we found no remarkable difference. Of note, CheckMate214 did not report the Karnofsky performance status score (PS) of patients, while the CLEAR trial presented a slightly higher number of patients with favorable MSKCC prognostic score and with better PS.

In the light of our results, at least three combination therapies seem to have the best therapeutic profile. In the clinical use, the selection of the most suitable treatment should be guided by the patient’s characteristics, particularly, the fitness of the subject to tolerate the adverse drug reactions (ADR), combined with a foresight on the best sequence of treatments if progression occurs [26]. Nivolumab + ipilimumab is burdened by more intense immune-related ADRs, while lenvatinib + pembrolizumab is characterized by a higher incidence of grade >3 ADR (82.4% of patients) than other TKI + ICI combinations [27]. Although ADR may impact on health-related quality of life (HRQOL), a recent study observed that patients treated with pembrolizumab + lenvatinib have similar HRQOL score than patients receiving sunitinib [28]. It would be worthwhile to extend the comparison of HRQOL scores to the combination regimens, evaluated in the present study in terms of OS and PFS.

The use of these combination treatments in real practice may generate further evidence to guide treatment selection. For example, a recent study investigating the efficacy of nivolumab + ipilimumab vs. pembrolizumab + axitinib in real practice showed a reduced PFS compared with that reported in the RCTs [29].

## 5. Conclusions

In conclusion, the “Shiny method” permitted to generate valuable clinical results on new treatments for aRCC and to discuss their efficacy on the basis of original indirect comparisons. In addition, the “one to many approach”, recently developed by our group, will allow us to quickly update the results of the present analysis as new treatments or longer follow-up will become available [30].

## Figures and Tables

**Figure 1 cancers-15-02029-f001:**
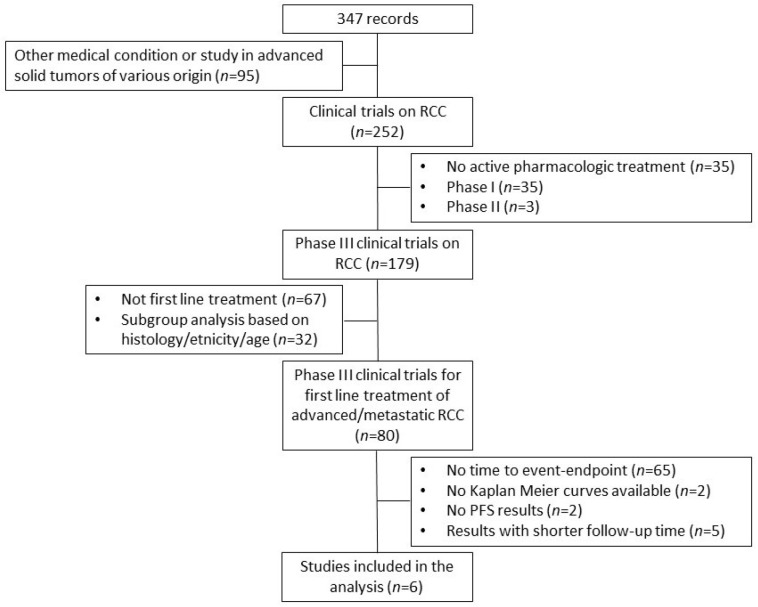
PRISMA flowchart of the process of trial selection. Renal cell carcinoma (RCC); Progression-free survival (PFS).

**Figure 2 cancers-15-02029-f002:**
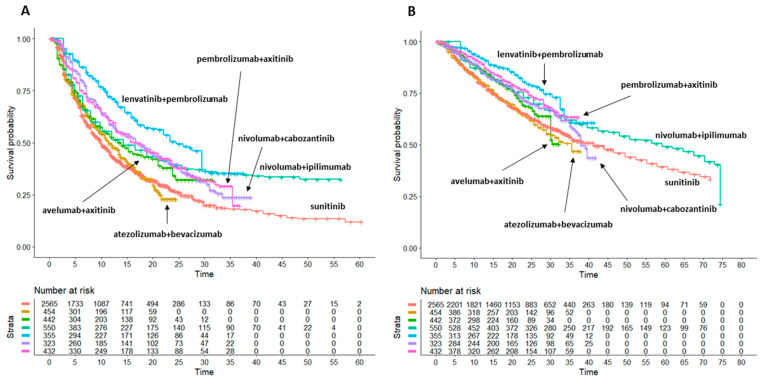
Application of the Shiny method to sunitinib monotherapy (*n* = 2565 from 6 trials; in red) and to 6 combination treatments: (a) atezolizumab 1200 mg + bevacizumab 15 mg/kg (*n* = 454; in gold); (b) avelumab 10 mg/kg + axitinib 5 mg (*n* = 442; in green); (c) nivolumab 3 mg/kg + ipilimumab 1 mg/kg (*n* = 550; in light green); (d) lenvatinib 20 mg + pembrolizumab 200 mg (*n* = 355; in light blue); (e) nivolumab 240 mg + cabozantinib 40 mg (*n* = 323; in purple); (f) and pembrolizumab 200 mg + axitinib 5 mg (*n* = 432; in pink); (g) sunitinib monotherapy 50 mg (*n* = 2565; in red). *n*: number of enrolled patients. End-point: (**A**) progression-free survival (PFS), (**B**) Overall survival (OS). Time in months.

**Figure 3 cancers-15-02029-f003:**
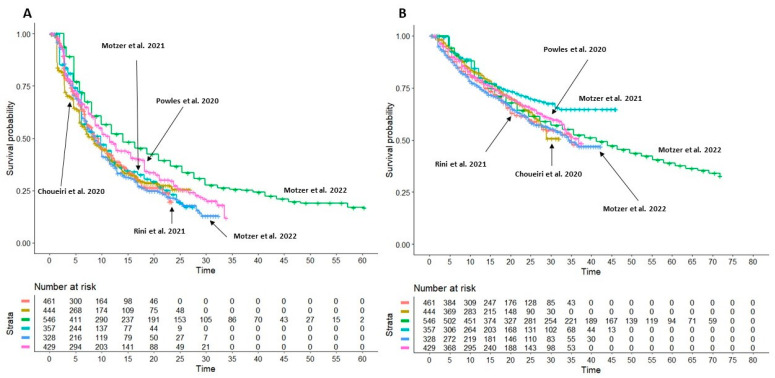
After reconstruction of individual patient data from control arms of included trials (sunitinib monotherapy 50 mg), the following Kaplan-Meier curves were generated from: Rini et al., 2021 (*n* = 461; in red [11]); Choueiri et al., 2020 (*n* = 444; in gold [12]); Motzer et al., 2022 (*n* = 546; in green [13]); Motzer et al., 2021 (*n* = 357; in light blue [14]); Motzer et al., 2022 (*n* = 328; in blue [15]); Powles et al., 2020 (*n* = 355; in purple [16]). (**A**) progression-free survival (PFS), (**B**) Overall survival (OS). Time in months.

**Figure 4 cancers-15-02029-f004:**
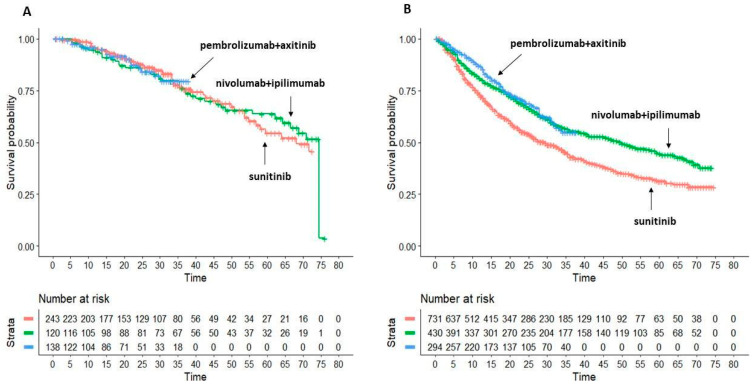
Overall survival Subgroups analysis. (**A**) In favourable-risk patients; application of the Shiny method to sunitinib monotherapy (*n* = 243 from 2 trials; in red) and to 2 combination treatments: (a) nivolumab 3 mg/kg + ipilimumab 1 mg/kg (*n* = 120; in green) and (b) pembrolizumab 200 mg + axitinib 5 mg (*n* = 138; in blue). (**B**) In intermediate-risk/poor-risk patients; Application of the Shiny method to sunitinib monotherapy (*n* = 731 from 2 trials; in red) and to 2 combination treatments: (a) nivolumab 3 mg/kg + ipilimumab 1 mg/kg (*n* = 430; in green) and (b) pembrolizumab 200 mg + axitinib 5 mg (*n* = 294; in blue). *n*: number of enrolled patients. End-point: overall survival (OS), Time in months.

**Table 1 cancers-15-02029-t001:** Main information about the 6 RCTs included in the analysis.

				PFS		OS	
#	Trial	Reference	Treatments under Comparison	Treatment Group (Events/ Patients)	Controls (Events/ Patients)	Treatment Group (Events/ Patients)	Controls (Events/ Patients)
RCC1	IMmotion151 trial (two-arm)	Rini et al. [11]	atezolizumab (1200 mg) + bevacizumab (15 mg/kg) intravenously, every 3 weeks;	283/454	309/461	194/454	192/461
			sunitinib (50 mg) orally once daily for 4 weeks (6 weeks cycle).				
RCC2	JAVELIN Renal 101 Trial (two-arm)	Choueiri et al. [12]	avelumab (10 mg/kg) intravenously, every 2 weeks + axitinib (5 mg) orally twice daily;	242/442	336/444	109/442	129/444
			sunitinib (50 mg) orally once daily for 4 weeks (6 weeks cycle).				
RCC3	CheckMate 214 Trial (two-arm)	Motzer et al. [13]	nivolumab (3 mg/kg) + ipilimumab (1 mg/kg) every 3 weeks for four cycles;	266/550	358/546	297/550	339/546
			sunitinib (50 mg) orally once daily for 4 weeks (6 weeks cycle).				
RCC4	CLEAR Trial (three-arm) §	Motzer et al. [14]	lenvatinib (20 mg) orally once daily + pembrolizumab (200 mg) intravenously, every 3 weeks;	160/355	205/357	80/355	105/357
			sunitinib (50 mg) orally once daily for 4 weeks (6 weeks cycle).				
RCC5	CheckMate 9ER Trial (two-arm)	Motzer et al. [15]	nivolumab (240 mg) intravenously, every 2 weeks + cabozantinib (40 mg) orally once daily;	206/323	223/328	121/323	150/328
			sunitinib (50 mg) orally once daily for 4 weeks (6 weeks cycle).				
RCC6	KEYNOTE-426 Trial (two-arm)	Powles et al. [16]	pembrolizumab (200 mg) intravenously every 3 weeks (35 cycles) + axitinib (5 mg) orally twice daily;	264/432	281/429	142/432	178/429
			sunitinib (50 mg) orally once daily for 4 weeks (6 weeks cycle).				

All values of event number were explicitly reported in the original trials or relative Appendix A; events were then calculated as the difference of the total number of patients minus the total number of censored cases. § CLEAR included an arm treated with lenvatinib (18 mg orally once daily) + everolimus (5 mg orally once daily), which has not been included in our analysis.

**Table 2 cancers-15-02029-t002:** Results of statistical analysis of indirect treatment comparison of PFS between combination regimens.

	HR	Lower 0.95	Upper 0.95
atezolizumab + bevacizumab vs. avelumab + axitinib	1.223	1.015	1.474
atezolizumab + bevacizumab vs. nivolumab + ipilimumab	1.462	1.227	1.743
atezolizumab + bevacizumab vs. pembrolizumab + lenvatinib	1.981	1.611	2.435
atezolizumab + bevacizumab vs. nivolumab + cabozantinib	1.415	1.163	1.721
atezolizumab + bevacizumab vs. pembrolizumab + axitinib	1.401	1.165	1.685
avelumab + axitinib vs. atezolizumab + bevacizumab	0.817	0.678	0.985
avelumab + axitinib vs. nivolumab + ipilimumab	1.195	0.998	1.432
avelumab + axitinib vs. pembrolizumab + lenvatinib	1.619	1.311	1.999
avelumab + axitinib vs. nivolumab + cabozantinib	1.156	0.946	1.413
avelumab + axitinib vs. pembrolizumab + axitinib	1.145	0.948	1.384
nivolumab + ipilimumab vs. atezolizumab + bevacizumab	0.684	0.574	0.815
nivolumab + ipilimumab vs. avelumab + axitinib	0.837	0.698	1.002
nivolumab + ipilimumab vs. pembrolizumab + lenvatinib	1.355	1.108	1.657
nivolumab + ipilimumab vs. nivolumab + cabozantinib	0.968	0.800	1.170
nivolumab + ipilimumab vs. pembrolizumab + axitinib	0.958	0.802	1.145
pembrolizumab + lenvatinib vs. atezolizumab + bevacizumab	0.505	0.411	0.621
pembrolizumab + lenvatinib vs. avelumab + axitinib	0.618	0.500	0.763
pembrolizumab + lenvatinib vs. nivolumab + ipilimumab	0.738	0.604	0.903
pembrolizumab + lenvatinib vs. nivolumab + cabozantinib	0.714	0.574	0.889
pembrolizumab + lenvatinib vs. pembrolizumab + axitinib	0.707	0.574	0.872
nivolumab + cabozantinib vs. atezolizumab + bevacizumab	0.707	0.581	0.860
nivolumab + cabozantinib vs. avelumab + axitinib	0.865	0.708	1.057
nivolumab + cabozantinib vs. nivolumab + ipilimumab	1.033	0.854	1.250
nivolumab + cabozantinib vs. pembrolizumab + lenvatinib	1.400	1.125	1.743
nivolumab + cabozantinib vs. pembrolizumab + axitinib	0.990	0.812	1.208
pembrolizumab + axitinib vs. atezolizumab + bevacizumab	0.714	0.594	0.858
pembrolizumab + axitinib vs. avelumab + axitinib	0.873	0.723	1.055
pembrolizumab + axitinib vs. nivolumab + ipilimumab	1.043	0.873	1.247
pembrolizumab + axitinib vs. pembrolizumab + lenvatinib	1.414	1.147	1.742
pembrolizumab + axitinib vs. nivolumab + cabozantinib	1.010	0.828	1.231

**Table 3 cancers-15-02029-t003:** Results of statistical analysis of indirect treatment comparison of OS between combination regimens.

	HR	Lower 0.95	Upper 0.95
atezolizumab + bevacizumab vs. avelumab + axitinib	1.263	0.97	1.645
atezolizumab + bevacizumab vs. nivolumab + ipilimumab	1.474	1.181	1.840
atezolizumab + bevacizumab vs. pembrolizumab + lenvatinib	1.888	1.412	2.523
atezolizumab + bevacizumab vs. nivolumab + cabozantinib	1.270	0.976	1.654
atezolizumab + bevacizumab vs. pembrolizumab + axitinib	1.639	1.256	2.138
avelumab + axitinib vs. atezolizumab + bevacizumab	0.791	0.608	1.030
avelumab + axitinib vs. nivolumab + ipilimumab	1.167	0.909	1.498
avelumab + axitinib vs. pembrolizumab + lenvatinib	1.494	1.093	2.042
avelumab + axitinib vs. nivolumab + cabozantinib	1.005	0.754	1.341
avelumab + axitinib vs. pembrolizumab + axitinib	1.297	0.970	1.734
nivolumab + ipilimumab vs. atezolizumab + bevacizumab	0.678	0.543	0.847
nivolumab + ipilimumab vs. avelumab + axitinib	0.857	0.667	1.1
nivolumab + ipilimumab vs. pembrolizumab + lenvatinib	1.28	0.970	1.690
nivolumab + ipilimumab vs. nivolumab + cabozantinib	0.862	0.671	1.106
nivolumab + ipilimumab vs. pembrolizumab + axitinib	1.112	0.864	1.431
pembrolizumab + lenvatinib vs. atezolizumab + bevacizumab	0.530	0.396	0.708
pembrolizumab + lenvatinib vs. avelumab + axitinib	0.669	0.490	0.915
pembrolizumab + lenvatinib vs. nivolumab + ipilimumab	0.781	0.592	1.031
pembrolizumab + lenvatinib vs. nivolumab + cabozantinib	0.673	0.492	0.920
pembrolizumab + lenvatinib vs. pembrolizumab + axitinib	0.868	0.634	1.189
nivolumab + cabozantinib vs. atezolizumab + bevacizumab	0.787	0.605	1.025
nivolumab + cabozantinib vs. avelumab + axitinib	0.995	0.746	1.327
nivolumab + cabozantinib vs. nivolumab + ipilimumab	1.161	0.904	1.490
nivolumab + cabozantinib vs. pembrolizumab + lenvatinib	1.486	1.087	2.031
nivolumab + cabozantinib vs. pembrolizumab + axitinib	1.29	0.965	1.724
pembrolizumab + axitinib vs. atezolizumab + bevacizumab	0.61	0.468	0.796
pembrolizumab + axitinib vs. avelumab + axitinib	0.771	0.577	1.03
pembrolizumab + axitinib vs. nivolumab + ipilimumab	0.9	0.699	1.158
pembrolizumab + axitinib vs. pembrolizumab + lenvatinib	1.152	0.841	1.577
pembrolizumab + axitinib vs. nivolumab + cabozantinib	0.775	0.58	1.036

## Data Availability

The data presented in this study are available in the article and in the Appendix A.

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
