# Peer review of "Progression-Free and Overall Survival of First-Line Treatments for Advanced Renal Cell Carcinoma: Indirect Comparison of Six Combination Regimens"

_cancers, 2023, doi:10.3390/cancers15072029_

Round 1

Reviewer 1 Report

The authors performed a systematic review of the literature and used the “Shiny Method”, to compare, indirectly, the efficacy, in terms of progression-free survival and overall survival, of the main ICI-based combination regimens for aRCC. I applaud the Authors for the attempt to produce new evidence on advanced renal cell carcinoma, a heterogeneous entity of which treatment landmarks still lack clear and definitive indications.

Nevertheless, some aspects need clarification by the authors.

-       The title should change because did not follow the current literature to report a systematic review and it is not adherent to the PRISMA statements.

-       The “Shiny method” is an innovative tool, not really known by most of the scientific community. They should expand the discussion of this tool throughout the whole paper. I suggest to include a brief explanation of the rationale for its use in the Introduction and adding a few words about how the software works in the Materials and methods. Furthermore, reporting some references from the current literature about results and evidence collected using this tool could enhance the quality of the Discussion.

-       The strength of the paper would have been improved if the authors followed PRISMA guidelines on systematic review. (e.g. registering the protocol on PROSPERO or adding risk of bias assessment and moreover reporting if a fixed or a random effect was used in the statistical analysis. Furthermore, it isn’t clear how the software “CADTH” have helped to perform indirect treatment comparison. I suggest the authors to explain clearly the use of this software in the statistical analysis.

-       Results are poorly reported in the manuscript. The details of which data have been extracted from each study are lacking and median and CI are only listed in the text. HR is not reported for every head-to-head comparison in the main text. I would suggest to report the HRs of each comparison in a table included in the main body, even better with a forest plot, being the main objective of the study.

-       All HR, PFS and OS mentioned in the Results, are different from the original RCTs data analysed (e.g. pembrolizumab+lenvatinib: median PFS 24.52 months (95%CI 22.02 to 29.5) vs the crude median PFS of ; 23.9 months (95% CI (20.8–27.7). It is not clear the advantage in clinical practice to change a real measure in an ideal and indirect one. The only pooled data are from sunitinib. All the others are not cumulative because are single drug combinations.

The explanation of the added value of the “Shiny method” seems to be too articulated in the Discussion. It would make the whole exposition easier to follow for the reader if the authors try to simplify this paragraph, focusing on the differences between a network -  meta-analysis and one performed with the “Shiny method” or maybe moving some sentences inherent to the statistical method to the relative paragraph.

Author Response

AUTHORS’ RESPONSE

We thank the Reviewers for their positive evaluation of our manuscript and for their helpful suggestions to improve it. In the revised version of our paper, we have addressed all the points raised by the Reviewers. The parts highlighted in yellow in this response have been included as such in the revised manuscript and in the manuscript we highlighted the text deleted in red type. Furthermore, the bibliography has been updated with all the articles cited in these responses.

In reply to Reviewer #1

Reviewer's requests:

“The title should change because it did not follow the current literature to report a systematic review and it is not adherent to the PRISMA statements”.

Response to the Reviewer:

Our study was not intended to be a systematic review on first-line treatments for advanced RCC. The original title of our paper was incorrect, and it has been rephrased as follows: “Progression-free and overall survival of first-line treatments for advanced renal cell carcinoma: indirect comparison of 6 combination regimens”

Reviewer's requests:

“The “Shiny method” is an innovative tool, not really known by most of the scientific community. They should expand the discussion of this tool throughout the whole paper. I suggest to include a brief explanation of the rationale for its use in the Introduction and adding a few words about how the software works in the Materials and methods. Furthermore, reporting some references from the current literature about results and evidence collected using this tool could enhance the quality of the Discussion”.

Response to the Reviewer:

We thank the Reviewer for this comment and we have added the requested explanation in the text.

In the Introduction: “The Shiny method, known also as IPDfromKM, is an innovative tool of survival analysis in which a software based on artificial intelligence reconstructs individual patient data. These reconstructed patients represent a new form of original clinical material; in particular, these reconstructed patients are suitable to perform indirect comparisons and  consequently determine the place in therapy of individual agents. In recent papers, we described the method and numerous experiences of application, especially in the field of oncology [9]. These analyses are helpful in providing an overview of new treatments, ranking their effectiveness based on indirect comparisons, and assessing equivalence from a regulatory viewpoint.”

Our last study recently published has been added to the Reference list:

  1. Messori, A.; Damuzzo, V.; Rivano, M.; Cancanelli, L.; Di Spazio, L.; Ossato, A.; Chiumente, M.; Mengato, D. Application of the IPDfromKM-Shiny Method to Compare the Efficacy of Novel Treatments Aimed at the Same Disease Condition: A Report of 14 Analyses. Cancers 2023, 15, 1633, doi:10.3390/cancers15061633.

In the materials and methods we have specified how the software generates the reconstructed dataset. The information needed as input in the software had already been described.

“The reconstructed data of the patients are stored in archives containing the following information: a) date of enrollment and last follow-up; the observation period of each patient results from the difference between these two dates. b) patient outcome at the last follow-up date (alive, dead or censored ).”

In the Discussion, we inserted a paragraph citing other research groups that have used the Shiny method in different fields of medicine. “In recent years, the Shiny method has increasingly been used not only in oncology but also in other areas of of therapeutics such as medical devices [20], surgery [21,22] and also in patients with COVID-19 [23]. However, Oncology and Oncoematology remain the two main areas of application, where the Shiny method  represents a quite simple alternative to network meta-analysis [8,9,24,25].”

Shiny investigations recently published have been added to the Reference list:

  1. Dimagli, A.; Cancelli, G.; Soletti, G.J.; Perezgrovas, O.R.; Chadow, D.; Rahouma, M.; Girardi, L.; Gaudino, M. Percutaneous coronary intervention versus repeat surgical revascularization in patients with prior coronary artery bypass grafting: A systematic review and meta-analysis. JTCVS Open. 2022, 12, 177-191, doi:10.1016/j.xjon.2022.10.006.
  2. Leung, Y.Y.R.; Bera, K.; Urriza, R. D.; Dardik, A.; Mas, J.L.; Simonte, G.; Rerkasem, K.; Howard, D.P.J. Safety of Carotid Endarterectomy for Symptomatic Stenosis by Age: Meta-Analysis With Individual Patient Data. Stroke. 2023, 54, 457-467, doi: 10.1161/STROKEAHA.122.040819.
  3. Magouliotis, D.E.; Zotos, P.A.; Karamolegkou, A.P.; Tatsios, E.; Spiliopoulos, K.; Athanasiou, T. Long-Term Survival after Extended Sleeve Lobectomy (ESL) for Central Non-Small Cell Lung Cancer (NSCLC): A Meta-Analysis with Reconstructed Time-to-Event Data. J Clin Med. 2022, 12, 204, doi: 10.3390/jcm12010204.
  4. Tan, B.K.J.; Han, R.; Zhao, J.J.; Tan, N.K.W.; Quah, E.S.H.; Tan, C.J.; Chan, Y.H.; Teo, N.W.Y.; Charn, T.C.; See, A.; et al. Prognosis and persistence of smell and taste dysfunction in patients with covid-19: meta-analysis with parametric cure modelling of recovery curves. BMJ. 2022, 27, 378:e069503, doi: 10.1136/bmj-2021-069503. Erratum in: BMJ. 2022, 9, 378:o1939.
  5. Yap, D.W.T.; Leone, A.G.; Wong, N.Z.H.; Zhao, J.J.; Tey, J.C.S.; Sundar, R.; Pietrantonio, F. Effectiveness of Immune Checkpoint Inhibitors in Patients With Advanced Esophageal Squamous Cell Carcinoma: A Meta-analysis Including Low PD-L1 Subgroups. JAMA Oncol. 2023, 9, 215-224, doi: 10.1001/jamaoncol.2022.5816.
  6. Fong, K.Y.; Zhao, J.J.; Sultana, R.; Lee, J.J.X.; Lee, S.Y.; Chan, S.L.; Yau, T.; Tai, D.W.M.; Sundar, R.; Too, C.W. First-Line Systemic Therapies for Advanced Hepatocellular Carcinoma: A Systematic Review and Patient-Level Network Meta-Analysis. Liver Cancer. 2022, 12, 7-18, doi: 10.1159/000526639.

Reviewer's requests:

“The strength of the paper would have been improved if the authors followed PRISMA guidelines on systematic review (e.g. registering the protocol on PROSPERO or adding risk of bias assessment and moreover reporting if a fixed or a random effect was used in the statistical analysis. Furthermore, it isn’t clear how the software “CADTH” have helped to perform indirect treatment comparison. I suggest the authors to explain clearly the use of this software in the statistical analysis”.

Response to the Reviewer:

We thank the reviewer for this suggestion. We confirm that the aim of this study was to perform a series of indirect comparisons using reconstructed patient data and not to perform a systematic review. Nevertheless, in our revised manuscript we included a detailed description of the literature search to document how the included trials were selected. Therefore, a PRISMA diagram is reported in our revised paper (including the number of eligible records, those included and excluded, and so on).
In our view, even though the number of Shiny articles has grown rapidly in the past year, this method should still be considered an investigational type of research, that still needs further standardisation. For example, while the generation of reconstructed patient data is a mandatory step in every Shiny research, other components, which are not considered mandatory, are however increasingly being recommended (e.g. the PRISMA diagram as mentioned above or the mandatory requirement to know the number of events and to avoid running the IPDfromKM software when this information is unavailable). Other components of the Shiny method require further methodological advancements, e.g. the methods to assess heterogeneity or the separate analysis that compares all the control groups with one another. In this framework, while the analysis of outcomes through the Shiny method is being standardised more and more, the assessment of safety remains an important weakness of the method,  which is unlikely to be solved in a short time.

Regarding the estimation of a hazard ratio through the CADTH software, the question is quite simple, but requires a long explanation. For example, if the objective of a time-to-event analysis is to compare three treatments (A, B, C) with one another, but only two head-to-head “real” trials” have been conducted (e.g. A vs C, and B vs C), the comparison of A vs B must be necessarily performed indirectly. The Shiny method places its analysis in a quite unreal framework in which the three patient cohorts treated with A, B, and C, respectively, are modelled altogether in a common framework and the distinction between real trials and  hypothetical trials goes lost. In other words, the three cohorts of reconstructed patients are modelled at exactly the same level, and so one can indifferently assume that all the reconstructed patients can be managed as real patients or, on the contrary, as unreal patients (which is absolutely the same in operational terms). On the side of statistical testing, a real framework can make the distinction between comparisons made earlier (typically those involving real trials) and comparisons made later (or “post-hoc”, i.e. those involving indirect comparisons in the absence of a real trial). By contrast, in the Shiny framework, the different cohorts are modelled within a common context, and no distinction is made between comparisons made earlier or later. Hence, the results of all comparisons are “stable” and do not depend on the nature of the comparisons or its “timing”.

In the analysis reported in our initial paper, we employed the Cox model firstly to  compare each of the 5 treatments with sunitinib. Then, to compare each of the 5 treatments  with the other treatments  (excluding sunitinib), we decided to use the CADTH software only because of  this test is very easy-to-use and because this would have been the choice in a real framework. However,  actually there was no need to use the CADTH software, because all indirect comparisons (in all head-to-head-combinations) can also be performed using the Cox model under the R-platform. The only requirement is to re-run the Cox model several times; in doing so, we have simply added a few lines of code in the R-script, so that the same script is run several times, i.e. as many times as the number of indirect comparisons. As a consequence of this choice, the CADTH software (and all citations of this software) have been removed from our revised paper. We have only explained the choice to run several times the R-code of the Cox model, by adding the following sentence: “Since numerous indirect comparisons were required by our analysis, the code under the R-platform that implements the Cox test was run as many times as the number of indirect  comparisons”. However, we ask Reviewer #1 if this sentence is useful or unnecessary.
As a confirmation of the explanation described above, when we have recomputed these HRs using additional R-code as a replacement of the CADTH software, we have obtained exactly the same HR values within the precision of 4 decimal digits. 

Reviewer's requests:

“Results are poorly reported in the manuscript. The details of which data have been extracted from each study are lacking and median and CI are only listed in the text. HR is not reported for every head-to-head comparison in the main text. I would suggest to report the HRs of each comparison in a table included in the main body, even better with a forest plot, being the main objective of the study”.

Response to the Reviewer:

In section 2.2 of materials and methods, we described which parameters were extracted from each study. These are the digitalized Kaplan-Meier curves along with the number of events and the number of patients at risk for active treatment and control arms of each trial. These details are reported in Table 1 both for PFS and OS.

In Table S1, we reported, for each trial, the medians (with 95%CI) of active treatments and sunitinib arm, and all HRs with 95%CI vs sunitinib for both OS and PFS.

HRs and 95%CI of each head-to-head comparison are reported in Table S3, but we decided, upon suggestion of Reviewer #1, to move this table in the Results section as Table 2.

We added a forest plot for the HR of PFS and OS of the six trials using pooled sunitinib patients as a control group to calculate the values of HRs (with 95%CI) shown in Figure S1. If requested, we can include in the Supplementary material six forest plots depicting HR and 95%CI for the 5 active treatments vs the sixth one. Still, we believe the multi-curve Kaplan-Meier curve depicted in Figure 2 is more effective in depicting the trend of survival of each treatment compared with one another.

Regarding the use of forest plots in a survival analysis, one drawback is that the forest plot ignores the duration of follow-up (and implicitly assumes that all included studies have had the same follow-up length, which is untrue). In this respect, one important advantage of the Shiny method is that the follow-up length of all included studies is taken into account.

All in all, our preference would be to limit the number of forest plots, because the assumption of forest plots negates the basis itself of the Shiny method.

Reviewer's requests:

“All HR, PFS and OS mentioned in the Results, are different from the original RCTs data analysed (e.g. pembrolizumab+lenvatinib: median PFS 24.52 months (95%CI 22.02 to 29.5) vs the crude median PFS of ; 23.9 months (95% CI (20.8–27.7). It is not clear the advantage in clinical practice to change a real measure in an ideal and indirect one. The only pooled data are from sunitinib. All the others are not cumulative because are single drug combinations”.

Response to the Reviewer:

The use of reconstructed patient data determines several advantages compared to real measures. First, it permits to summarise results in a single Kaplan-Meier graph where each active treatment constitutes an arm of the graph. This yields a visual comparison at a glance between the treatments. Second, reconstruction of patients data permits to pool all patients treated with sunitinib into a single control group where the effect of the selection bias related to individual studies is minimised. There are numerous advantages resulting from the presence of a pooled control group (e.g. in estimating the magnitude of the effect of each treatment against a common control group of patients receiving sunitinib).

Several studies have demonstrated that the HR estimated from reconstructed data are very similar to those reported in the original trial [7]. Other studies on this issue are in progress.

Nevertheless, some comments are needed about this observation made by Reviewer #1.

In the first place, the article needs consistency, and so the main two choices are either to report “reconstructed” values of HR throughout the entire manuscript or to report original values of HR. The first option is mandatory because our article contains a very large number of hypothetical patient groups (e.g. the pooled control group generated from the control groups of the 6 trials) for which the original values of HR are unavailable. Consequently, the option of reporting only the original values of HR is unfeasible.

On the other hand, there are a number of patient groups for which both the reconstructed values of HR were available as well as the original values of HR published in the trials. In these cases, we have given priority to the reconstructed HR for reasons of internal consistency within the article.

Reviewer's requests:

“The explanation of the added value of the “Shiny method” seems to be too articulated in the Discussion. It would make the whole exposition easier to follow for the reader if the authors try to simplify this paragraph, focusing on the differences between a network - meta-analysis and one performed with the “Shiny method” or maybe moving some sentences inherent to the statistical method to the relative paragraph”.

Response to the Reviewer:

We have simplified the paragraph according to the Reviewer’s suggestion. Thank you for your comment. We have also corrected all points indicated by the Reviewer.

Reviewer 2 Report

Dear all,

After reviewing this paper "Progression-free and overall survival after first-line treatments for advanced renal cell carcinoma: indirect comparison of 6 combination regimens".
I found this topic innovative and ethical and aimed at reconstructing individual patient data to identify the best combination regimens in aRCC after considering patients' heterogeneity.

In my opinion, this article is accepted for publication after minor revisions:

In fig. 2 and 3: Change the colour of the line reporting lenvatinib + pembrolizumab (in light blue).

line 203-204: Explain how
the presence of heterogeneity is driven mainly by the nivolumab+ipilimumab RCT control arm.

In the conclusion: Can you comment on the efficiency of this application by correlating the outcome of this application (the best 3 combinations mode of action) to the different patient profiles/prognoses?

Author Response

AUTHORS’ RESPONSE

We thank the Reviewers for their positive evaluation of our manuscript and for their helpful suggestions to improve it. In the revised version of our paper, we have addressed all the points raised by the Reviewers. The parts highlighted in yellow in this response have been included as such in the revised manuscript and in the manuscript we highlighted the text deleted in red type. Furthermore, the bibliography has been updated with all the articles cited in these responses.

In reply to Reviewer #2

Reviewer's requests:

line 203-204: Explain how the presence of heterogeneity is driven mainly by the nivolumab+ipilimumab RCT control arm.

Response to the Reviewer:

To demonstrate that the presence of heterogeneity across the control groups is driven mainly by the nivolumab+ipilimumab control arm, in a separate analysis we have re-analysed the likelihood ratio test for both PFS and OS after excluding the nivolumab+ipilimumab control arm.

We added in the 3.4. paragraph as follow: “The presence of heterogeneity is driven mainly by the nivolumab+ipilimumab and lenvatinib+pembrolizumab RCT control arm respectively in the PFS and OS analysis.

In the case of PFS, there is no longer a significant heterogeneity  when the nivolumab+ipilimumab control group is left out (likelihood ratio test= 8.86 on 4 df, p=0.06); this demonstrates that in this case the heterogeneity is driven mainly by the this  control arm.

In contrast, in the case of  OS analysis, there is still significant heterogeneity after excluding  nivolumab+ipilimumab control group (likelihood ratio test=14.6 on 4 df, p=0.006). In this case, lenvatinib+pembrolizumab control arm performed significantly better than the others and so the heterogeneity was driven by this curve in the OS analysis. As expected, the likelihood ratio test after excluding lenvatinib+pembrolizumab control group was not significant (likelihood ratio test= 3.56 on 4 df,  p=0.5).”

To identify the main determinants of this heterogeneity, in the first place we compared the patients characteristics of the trials according to the information presented in Table 1. No striking differences emerged from this comparison. The only difference is that, in the lenvatinib+pembrolizumab group, the patients had a slightly better performance status than the patients included in the other trials. Since the performance status is an important predictor of survival, this might explain in part why the patients of this trial treated with sunitinib had a better OS compared to the patients treated with sunitinib in the other trials.

Reviewer's requests:

In the conclusion: Can you comment on the efficiency of this application by correlating the outcome of this application (the best 3 combinations mode of action) to the different patient profiles/prognoses?

Response to the Reviewer:

As reported in the discussion of the study, in patients with favourable risk, the head-to-head comparisons through “Shiny method” did not show any difference between sunitinib and the two combination treatments. In contrast, in the intermediate-poor risk group, nivolumab+ipilimumab (as well as pembrolizumab+axitinib) performed significantly better than sunitinib.

We have added this consideration in the conclusion.

Minor comments:

In fig. 2 and 3: Change the colour of the line reporting lenvatinib + pembrolizumab (in light blue).

Response to the Reviewer:

As requested, for a better comprehension and consistency between different figures, we have modified the figure caption 2, 3 and 4.

Round 2

Reviewer 1 Report

I Accept the revised manuscript